# A Rare Encounter: Gastric Ulcer Penetration into the Splenic Hilum Presenting with Upper Gastrointestinal Bleeding and a Massive Splenic Haematoma—Case Report and Literature Review

**DOI:** 10.3390/diagnostics15050617

**Published:** 2025-03-04

**Authors:** Ioana-Irina Rezuș, Vasile-Claudiu Mihai, Diana Elena Floria, Andrei Olteanu, Vlad Ionut Vlasceanu, Radu Petru Soroceanu, Alin Constantin Pinzariu, Brigitta Teutsch, Sergiu Tudose-Timofeiov

**Affiliations:** 1Department of Radiology, “Grigore T. Popa” University of Medicine and Pharmacy, 700115 Iași, Romania; rezus_ioana-irina@d.umfiasi.ro; 2Radiology Clinic, “St. Spiridon” County Clinical Emergency Hospital, 700111 Iași, Romania; 3Centre for Translational Medicine, Semmelweis University, 1085 Budapest, Hungary; iovdiana95@gmail.com (D.E.F.); teutschbrigitta@gmail.com (B.T.); 41st Internal Medical Department, “Grigore T. Popa” University of Medicine and Pharmacy, 700115 Iași, Romania; olteanuandrei@yahoo.com; 5Institute of Gastroenterology and Hepatology, “St. Spiridon” County Clinical Emergency Hospital, 700111 Iași, Romania; 6Department of Surgery I, ”Grigore T. Popa” University of Medicine and Pharmacy, 700115 Iași, Romania; vlad-ionut.vlasceanu@umfiasi.ro (V.I.V.); stimof@yahoo.com (S.T.-T.); 7Department of Surgery, “St. Spiridon” County Clinical Emergency Hospital, 700111 Iași, Romania; 8Department of Morpho-Functional Sciences II, “Grigore T. Popa” University of Medicine and Pharmacy, 700115 Iași, Romania; alin.pinzariu@umfiasi.ro; 9Department of Radiology, Medical Imaging Centre, Semmelweis University, 1082 Budapest, Hungary

**Keywords:** peptic ulcer disease, penetrating ulcer, atypical gastric resection, distal splenopancreatectomy, haemorrhage, haematoma

## Abstract

**Background:** Despite advancements in prevention and treatment, peptic ulcer disease (PUD) remains a public health burden, with potentially high mortality rates when not managed properly. Recent studies indicate bleeding as the most prevalent complication, followed by perforation or penetration into adjacent organs and pyloric obstruction. In rare cases, posterior wall or greater curvature ulcers of the stomach can penetrate, leading to splenic artery pseudoaneurysms. With nonspecific symptoms and low incidence, it is highly important that these entities are not overlooked in the diagnosis of patients with upper gastrointestinal bleeding. **Case Report:** We present the case of a 44-year-old patient presenting for upper abdominal pain and haematemesis while being haemodynamically stable. Emergency ultrasound described a dysmorphic spleen, with a transonic image with a Doppler signal in the splenic hilum. Upper gastrointestinal tract endoscopy detected a blood-filled stomach, without the possibility of identifying the bleeding source. The CT scan revealed active bleeding with peri splenic haematoma. Intraoperatively, a posterior gastric wall penetration into the spleen was identified, and an atypical gastric resection and caudal splenopancreatectomy were performed. The postoperative course was marked by the identification of a staple line leak in the upper pole of the stomach, which was treated conservatively, with a favourable outcome, and the patient was discharged after two weeks. **Conclusions:** Upper gastrointestinal tract haemorrhage needs fast intervention and suitable management. The multidisciplinary team plays a key role in identifying and treating rare causes such as penetration into the splenic hilum.

## 1. Introduction

Peptic ulcer disease (PUD) still remains a significant global health concern due to its potential complications, with a lifetime prevalence estimated at 5–10% in the general population and an annual incidence of 0.1–0.3% [1,2]. Despite significant advancements in understanding its pathophysiology and introducing effective treatment strategies, including proton pump inhibitors and *Helicobacter pylori* eradication, complications are still encountered in 10–20% of affected patients [3]. Despite a decline in the incidence, need for hospitalisation, and mortality rates associated with PUD, severe complications such as bleeding, perforation, and penetration into adjacent organs persist [2,3].

Bleeding is the most common complication, followed by perforation, which occurs at a ratio of approximately 1:6. Though less frequent, perforations constitute the leading indication for emergency surgery in cases of complicated ulcer disease and account for most deaths, with significant short-term mortality estimated to be around 10% to 30% [4,5]. While most perforations occur freely into the peritoneal cavity, penetration into adjacent structures such as the liver, pancreas, or spleen is also possible. These are usually associated with atypical clinical presentations, leading to considerable diagnostic challenges. Physical examination findings may be inconclusive, as signs of peritonitis can be minimal or absent in up to one-third of patients, particularly in cases with a contained or sealed leak [6]. Proper management of this condition is complex, involving multidisciplinary teams including surgeons, gastroenterologists, and radiologists.

This paper aims to describe a complicated gastric ulcer penetrating the splenic hilum, presenting with a voluminous peri splenic haematoma and subsequent upper gastrointestinal bleeding. By highlighting the diagnostic and therapeutic challenges posed by rare complications of gastric ulcers, this report aims to improve the recognition and management of such uncommon conditions.

## 2. Case Presentation

We present the case of a 44-year-old male who presented to the Emergency Department of a tertiary centre in North-Eastern Romania for upper abdominal pain and multiple episodes of haematemesis. The patient was haemodynamically stable upon presentation, with evidence of normocytic anaemia, indicated by a haemoglobin level of 8.8 g/dL. Regarding his past medical history, he was diagnosed with chronic pancreatitis, with multiple previous episodes of acute alcoholic pancreatitis. He had a history of chronic ethanol and tobacco use.

An abdominal ultrasound examination was performed in the Emergency Department, which described a highly altered spleen structure with hypoechoic areas. A round, well-defined transonic lesion measuring 27/30 mm with a partial Doppler signal was noticed in the splenic hilum. The supposition of a splenic artery (SA) pseudoaneurysm was raised.

The patient also underwent an upper gastrointestinal endoscopy in the Gastroenterology Department within 12 h of presentation in the Emergency Department. Upon examination, the stomach was filled with fresh blood and clots that could not be effectively washed or aspirated (Figure 1). A protrusion was observed on the posterior wall of the stomach along the greater curvature, suggesting extrinsic compression. However, the exact source of the bleeding could not be adequately identified. An abdominal computed tomography (CT) scan was further requested, which described findings of a possible splenic laceration with peri splenic haematoma and active bleeding. The aforementioned haematoma had no clear demarcation from the posterior gastric wall, and the splenic hilum was challenging to differentiate (Figure 2).

Given the need for immediate management, and without access to interventional radiology procedures, an exploratory laparotomy was performed. Intraoperatively, a voluminous tumoral mass was observed, involving the greater curvature of the stomach and splenic hilum (Figure 3). Due to anatomical changes and adherences caused by multiple episodes of acute pancreatitis, a decision was made to remove the mass en bloc. Thus, a longitudinal mechanical gastric resection along the greater curvature, associated with a distal splenopancreatectomy, was performed. The gastric staple line was reinforced with a continuous 3-0 absorbable running suture, and the caudal pancreatic stump was closed with an x-shaped suture. The histopathological examination of the resected block revealed a chronic peptic ulcer located on the posterior gastric wall, which perforated into the spleen. The immediate postoperative course of the patient was uneventful.

Five days following the surgical intervention, a chest X-ray showed a small unilateral left-sided pleural effusion, for which ultrasound-guided drainage was performed (600 mL of clear liquid). On microbiological analysis, the liquid tested positive for *Escherichia coli*, for which the patient was given antibiotics. Repeated abdominal ultrasounds found no evidence of intraperitoneal free fluid during the hospital stay. One week after surgery, another CT scan was performed, describing a left subphrenic collection suggestive of an abdominal staple line leak, near the angle of Hiss, which was successfully managed conservatively. The patient exhibited a good general condition, was afebrile, maintained digestive tolerance, had no signs of peritoneal irritation on palpation, and the drainage tubes were removed. The patient was discharged on the 14th postoperative day with recommendations to stop ethanol and tobacco use, to avoid intense physical effort, and to follow the vaccination protocol required after splenectomy.

## 3. Discussion

This case report describes a relatively rare but severe complication of PUD, with several particular characteristics. Firstly, the patient presented with a peptic ulcer involving the posterior gastric wall, an uncommon site accounting for only 5–8% of all gastroduodenal ulcers [7]. Left untreated, posterior perforation is a rare but possible complication [6,8]. One study cites records of only six similar cases over 12 years [9], with other working groups reporting even lower incidences [10]. Secondly, the patient had several risk factors for developing an SA pseudoaneurysm. A history of recurrent admissions for acute pancreatitis and a confirmed diagnosis of chronic pancreatitis places the patient at higher risk for vascular wall weakening due to the effects of pancreatic enzymes [11]. Moreover, enzymes resulting from a gastric perforation may add up to vascular wall damage, potentially resulting in the formation of an SA pseudoaneurysm [12,13].

Prompt and effective management of these cases in the Emergency Department is essential. Given the fact that symptoms are often nonspecific, a thorough understanding of this pathology is important [14]. We conducted a systematic search on MEDLINE (via Pubmed) on 7 December 2024, and found nine similar articles available in the literature. As illustrated in Table 1, diagnostic approaches do not rely on a single gold standard but instead require a minimum set of available techniques. Nowadays, CT angiography (CTA) is the most commonly used technique to diagnose SA pseudoaneurysms, with great sensitivity and specificity [15]. These typically present as localised outpouchings originating from the vessel’s lateral wall. Unlike active haemorrhage, pseudoaneurysms retain their form in delayed-phase imaging. In contrast, active bleeding is characterised by dynamic changes in the shape and intensity of contrast [11]. CT angiography plays a crucial role in the emergency management of patients presenting with signs of gastrointestinal bleeding. It has many advantages, such as locating the source of blood extravasation with an accuracy of almost 100% and providing information on any existing vascular anatomical variants [16].

Perforated gastric ulcers require immediate intervention. Given the location of the lesion and splenic involvement, an atypical gastric resection, along with a distal splenopancreatectomy, was performed. The extent of the gastrectomy and the decision to undertake an en bloc resection of adjacent organs depend on the damage to the spleen and/or pancreas. Notably, these surgical interventions have high mortality and morbidity rates, as well as significant postoperative risks [17,18]. Potential complications following surgery include metabolic alterations after gastrectomy [19], pancreatic fistula after partial pancreatectomy [20], and infections after splenectomy [21]. In addition to that, minimally invasive therapeutic options have become available in recent years. Percutaneous angioembolisation demonstrates good efficacy rates, ranging from 79% to 100%, with mortality lower than 20% [11]. Endovascular management consists of transcatheter embolisation with either coils or covered stents, with multiple available techniques of material insertion [22]. In our case, we acknowledge that performing a gastric resection combined with distal splenopancreatectomy is a highly demanding procedure, typically reserved for select cases. However, in this particular situation, the intraoperative findings dictated the necessity of an extended resection, and delaying the intervention could have resulted in significant morbidity or mortality. At the time of surgery, the available blood reserves were limited, and the patient presented with an ongoing active haemorrhage from the lesion. Furthermore, due to multiple episodes of acute pancreatitis, the local anatomy was significantly distorted, with dense adhesions between the spleen, pancreas, and lateral gastric wall. These adhesions rendered an isolated splenectomy infeasible, as the spleen was firmly attached to the gastric wall. Additionally, intraoperative assessment revealed a severely compromised spleen, structurally disorganised and undergoing enzymatic digestion due to prolonged exposure to corrosive gastric secretions. Given these findings, the only viable therapeutic option to achieve haemostasis and prevent further complications was the combined gastric resection and distal splenopancreatectomy.

The patient’s recovery progressed without major complications, and he was discharged two weeks after surgery. However, he developed a post-surgical pleural effusion, with pleural fluid testing positive for *Escherichia coli*. As multidrug resistance to this infection has decreased in our region [23], the pathogen was sensitive to a wide range of antibiotics, allowing the infection to be effectively managed without further challenges. Similarly, in other cases reported in the literature, patients typically show favourable evolution, with an average discharge time of 13 days following surgery [8,12,13,14,15,24,25,26].

This case is notable for the rare complication of a chronic gastric ulcer penetrating into the splenic hilum, leading to significant anatomical disruption and presenting with upper gastrointestinal bleeding. The patient’s history of chronic pancreatitis, ethanol, and tobacco use contributed to the development of the ulcer and associated complications. Diagnostic imaging revealed complex findings, including a splenic laceration, a poorly demarcated splenic hilum, and a peri splenic haematoma, which added to the challenge of identifying the exact source of bleeding preoperatively. The involvement of the greater curvature of the stomach and posterior gastric wall further emphasises the unique nature of the case.

## 4. Conclusions

This article reports a particular clinical case of a complicated gastric ulcer penetrating into the spleen, presenting with upper gastrointestinal bleeding and a voluminous pseudoaneurysm in the splenic hilum. This presentation underscores the complexity of managing rare complications of peptic ulcer disease and highlights the importance of prompt recognition and therapeutic intervention. A multidisciplinary team including surgeons, gastroenterologists, and radiologists is fundamental in the management of these patients, in order to improve clinical outcomes.

## Figures and Tables

**Figure 1 diagnostics-15-00617-f001:**
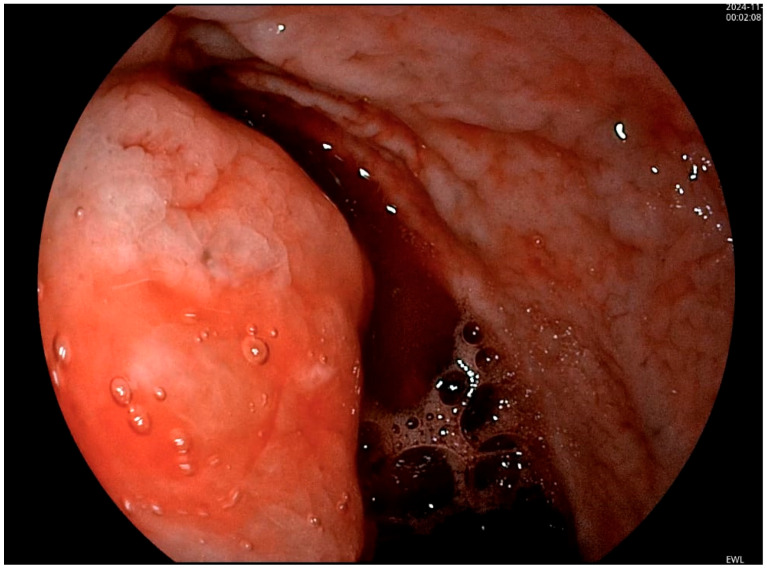
Endoscopic aspect highlighting the presence of fresh blood into the stomach, alongside a protrusion on the posterior wall and the greater curvature, suggesting extrinsic compression.

**Figure 2 diagnostics-15-00617-f002:**
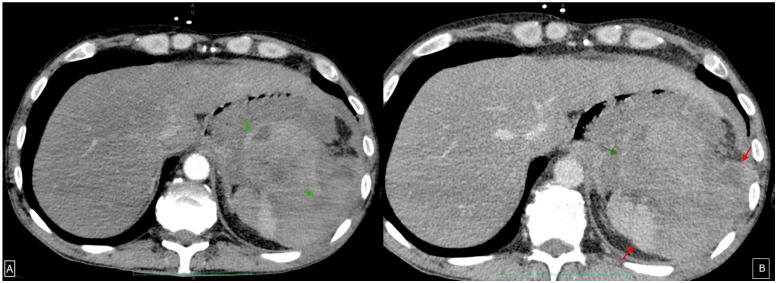
CECT in arterial (**A**) and venous phase (**B**) demonstrating a hyperdense collection (green arrows) located between the stomach anteriorly and the heterogeneously enhancing spleen (red arrows) posteriorly, with indiscernible margins.

**Figure 3 diagnostics-15-00617-f003:**
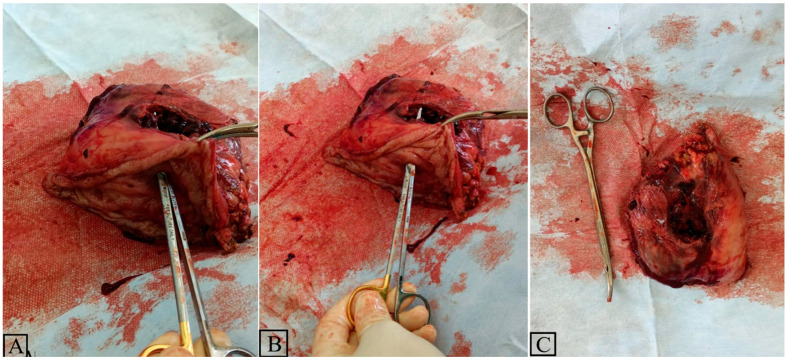
(**A**–**C**)—Intraoperative aspect of the spleen attached to the perforated posterior stomach wall. The scissors indicate the site of perforation.

**Table 1 diagnostics-15-00617-t001:** Summary of similar case reports available in the literature.

Article	Patient	Symptoms	Imaging	Treatment	Outcome
Syed et al., 2014 [15]	77 yo, female	Epigastric pain, haematemesis Haemodynamically stable	E: blood clots in the lumen of the stomach; A: 4 cm mid-SA pseudoaneurysm	TAE of the SA En bloc resection of SA pseudoaneurysm, distal pancreatectomy, partial gastrectomy, splenectomy	No complications Discharge fifth day
Cho et al., 2018 [16]	61 yo, male	Melena, dizziness Haemodynamically stable	E: blood in the stomach, gastric ulcer + an exposed vessel in the posterior wall of the gastric high body; CT: haematoma-filled stomach, splenic infarction, and pseudoaneurysm of the SA; A: pseudoaneurysm of SA	TAE of the SA (sandwich method) Gastrectomy	No complications Discharge eighth day
Sawicki et al., 2015 [17]	57 yo, male	Syncope preceded by severe abdominal pain Haemodynamically unstable	E: no source of bleeding; US: no abnormality; A: ruptured distal SA pseudoaneurysm on the posterior stomach wall	Bipolar ligation of the bleeding vessel, suture of gastric ulcer	Complications: ARDS, perigastric haematoma Discharge not mentioned
Garg et al., 2015 [18]	25 yo, male	Upper abdominal pain radiating to the back, vomiting Haemodynamically stable	US: multiple coarse calcifications in the pancreatic head, fat stranding, mild collection in the left anterior para-renal space; CT: destruction of the splenic parenchyma with protrusion of the remaining tissue into the stomach lumen	Emergency surgery	Not mentioned
Shidahara et al., 2022 [19]	33 yo, male	2-week haematemesis, impaired consciousness Haemodynamically unstable	CT: massive, enhanced fluid collection in the stomach E: fresh blood in the stomach, gastric ulcer on the posterior wall of the gastric body A: extravasation from a splenic pseudoaneurysm	TAE Emergency laparotomy—SA ligation, proximal gastrectomy (the double-tract method)	Post-op day 7—partial splenic infarction Discharge 15th day
Pasumarthy et al., 2009 [20]	55 yo, male	Abdominal pain, haematemesis, melena Haemodynamically unstable	CT: active contrast extravasation into the stomach from the SA A: area of active contrast extravasation from the mid portion of the SA beyond the dorsal pancreatic artery	TAE Total gastrectomy, splenectomy	No complications Discharge not mentioned
Varshney et al., 2014 [21]	38 yo, male	Abdominal pain, haematemesis, melena	E: large ulcer with the undermined edge along lesser curvature at the fundus, no active bleeding CT: SA pseudoaneurysm with multiple splenic infarcts	Primary ulcer repair, splenectomy	No complications Discharge eighth day
Nakata et al., 2022 [22]	72 yo, female	Melena Haemodynamically unstable	E: profuse posterior stomach wall haemorrhage CT: free air in the abdomen, a large amount of fluid in the stomach	REBOA Distal gastrectomy with Roux-en-Y reconstruction	No complications Discharge 29th day
Tessier et al., 2003 [23]	mean age 51.2 yo, five male, five female	Haematemesis (7) Haemodynamically unstable (2)	CT (5), A (3), E (1)—diagnosis of SA pseudoaneurysm	Splenectomy + distal pancreatectomy (4), splenectomy only (2), TAE (2), simple ligation (1)	Preoperative death (1) Pneumonia (3) Pancreatic ductal leak (1) Acute renal failure (1)

Years old (yo), upper endoscopy (E), diagnostic angiography (A), computed tomography (CT); transcatheter arterial coil embolisation (TAE), splenic artery (SA), ultrasound (US), resuscitative endovascular balloon occlusion of the aorta (REBOA).

## Data Availability

The original contributions presented in the study are included in the article; further inquiries can be directed to the corresponding author.

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
