# Peer review of "A Rare Encounter: Gastric Ulcer Penetration into the Splenic Hilum Presenting with Upper Gastrointestinal Bleeding and a Massive Splenic Haematoma—Case Report and Literature Review"

_diagnostics, 2025, doi:10.3390/diagnostics15050617_

Round 1
Reviewer 1 Report
Comments and Suggestions for Authors
The authors presented a case of successful treatment of a patient with a gastric ulcer penetrating into the spleen hilum presenting with upper gastrointestinal bleeding and a massive splenic haematoma.
Questions and comments:
1. Did the patient have clinical signs of bleeding from the upper gastrointestinal tract upon admission?
2. When did the patient undergo an upper gastrointestinal endoscopy since admission?
3. Why, when a splenic artery pseudoaneurysm was suspected according to abdominal ultrasound examination, was computed tomography angiography (angio CT) and/or magnetic resonance imaging (MRI) not performed?
4. If the indication for surgery was bleeding, why wasn't endovascular embolization of the splenic artery pseudoaneurysm applied, which was more appropriate? As I understand it, the gastric ulcer penetration was detected only during the operation.
5. I believe that performing gastric resection in combination with distal splenopancreatectomy in the situation described by the authors was very risky. In my opinion, such an intervention should have been performed only after stabilization of the patient and a thorough examination. I suggest that the authors provide a comprehensive explanation on this matter.
Author Response
Comment 1: Did the patient have clinical signs of bleeding from the upper gastrointestinal tract upon admission?
Response 1: Thank you for your question. Yes, the patient presented with upper abdominal pain and multiple episodes of hematemesis. We mentioned these aspects at the beginning of the 'Case Presentation' section.
Comment 2: When did the patient undergo an upper gastrointestinal endoscopy since admission?
Response 2: Thank you for this insightful question. The patient was assessed by upper gastrointestinal endoscopy within 12 hours of presentation in the Emergency Department. We added this information in the 'Case Presentation' section to make it clearer for our potential readers.
Comment 3: Why, when a splenic artery pseudoaneurysm was suspected according to abdominal ultrasound examination, was computed tomography angiography (angio CT) and/or magnetic resonance imaging (MRI) not performed?
Response 3: Thank you for this question. The patient presented to the Emergency Department with hematemesis, so an upper GI endoscopy was prioritized as the first diagnostic step to assess for potential sources of bleeding, as he was hemodynamically stable and fit for endoscopy. A CT scan of the abdomen and pelvis was performed afterwards to further evaluate overall abdominal pathology, not just vascular abnormalities.
Comment 4: If the indication for surgery was bleeding, why wasn't endovascular embolization of the splenic artery pseudoaneurysm applied, which was more appropriate? As I understand it, the gastric ulcer penetration was detected only during the operation.
Response 4: Thank you for your comment. Unfortunately, interventional radiology is not routinely available at our centre. Therefore, the optimal therapeutic option for our patient in the emergency setting was surgical treatment.
Comment 5: I believe that performing gastric resection in combination with distal splenopancreatectomy in the situation described by the authors was very risky. In my opinion, such an intervention should have been performed only after stabilization of the patient and a thorough examination. I suggest that the authors provide a comprehensive explanation on this matter.
Response 5: Thank you for your insightful observation and for giving us the opportunity to further clarify our surgical approach in this complex case. We acknowledge that performing a gastric resection combined with distal splenopancreatectomy is a highly demanding procedure, typically reserved for select cases. However, in this particular situation, the intraoperative findings dictated the necessity of an extended resection, and delaying the intervention could have resulted in significant morbidity or mortality. At the time of surgery, the available blood reserves were limited, and the patient presented with ongoing active hemorrhage from the lesion. Furthermore, due to multiple episodes of acute pancreatitis, the local anatomy was significantly distorted, with dense adhesions between the spleen, pancreas, and lateral gastric wall. These adhesions rendered an isolated splenectomy infeasible, as the spleen was firmly attached to the gastric wall. Additionally, intraoperative assessment revealed a severely compromised spleen, structurally disorganized and undergoing enzymatic digestion due to prolonged exposure to corrosive gastric secretions. Given these findings, the only viable therapeutic option to achieve hemostasis and prevent further complications was the combined gastric resection and distal splenopancreatectomy. We appreciate your valuable input and hope that this explanation provides a clear rationale for our surgical decision-making. Please let us know if further clarifications are required.
Reviewer 2 Report
Comments and Suggestions for Authors
Dear Editor,
Dear Author,
I read with great interest the manuscript entitled “A Rare Encounter: Gastric Ulcer Penetration into the Splenic Hilum Presenting with Upper Gastrointestinal Bleeding and a Massive Splenic Haematoma—Case Report and Literature Review” by RezuÈ™ et al.
This was a case report of gastric ulcer penetration into the splenic hilum presenting with NVUGIB, diagnosed through MDCTA and successfully treated surgically. A brief review on this topic is also given by the authors.
I consider the manuscript well written, well presented and relevant for the research context.
However, I have the following minor comment only:
- The crucial role of CT angiography in the diagnosis of selected cases of non-variceal upper GI bleeding, especially among rare sources and special situations (such as the presented case), should be highlighted in the discussion section and proper citations should be provided.
Author Response
Comment 1: The crucial role of CT angiography in the diagnosis of selected cases of non-variceal upper GI bleeding, especially among rare sources and special situations (such as the presented case), should be highlighted in the discussion section and proper citations should be provided.
Response 1: Thank you for this insightful suggestion! We agree with it, and we included a small paragraph in the Discussion section to highlight the advantages of CT angiography. However, we could not extend this topic further due to the strict word count limitation for case reports.
Round 2
Reviewer 1 Report
Comments and Suggestions for Authors
Please add the answers to comments 4 and 5 to the manuscript text.
Author Response
Comment 1: Please add the answers to comments 4 and 5 to the manuscript text.
Response 1: Thank you for your suggestion! We acted accordingly and added the answers to the text.